# Beating LM-cut with LM-cut:
# Quick Cutting and Practical Tie Breaking for the Precondition Choice Function

**Pascal Lauer** and **Maximilian Fickert**

Saarland University
Saarland Informatics Campus
Saarbrücken, Germany
s8palaue@stud.uni-saarland.de, fickert@cs.uni-saarland.de

## Abstract

LM-cut is one of the most popular heuristics in optimal planning that computes strong admissible estimates of the perfect delete relaxation heuristic $h^+$. The heuristic iteratively computes disjunctive action landmarks for the current state, reducing their action costs until no more landmarks with remaining action costs can be found. These landmarks are generated by finding cuts in the justification graph, which depends on a precondition choice function mapping each action to its most expensive precondition according to $h^{\max}$. This precondition is not necessarily unique, yet the performance of the heuristic heavily depends on this choice. We introduce and analyze several new tie breaking strategies for the precondition choice function, and evaluate their effectiveness on the IPC benchmarks. Furthermore, we suggest a modification to the computation of the cut, which trades a negligible loss in heuristic accuracy for a significant speedup of the LM-cut computation.

## Introduction

In optimal classical planning, strong admissible heuristics are desired to find solutions through heuristic search. One such heuristic is the landmarks cut heuristic $h^{\text{LM-cut}}$ (Helmert and Domshlak 2009), which provides high-quality estimates of the perfect delete relaxation heuristic $h^+$. While LM-cut has been surpassed by recent advancements on abstraction heuristics and cost partitionings (e.g. Franco et al. 2017; Seipp and Helmert 2018; Seipp, Keller, and Helmert 2020), it remains a popular heuristic that does not require any precomputation before search.

LM-cut is defined by iteratively computing $h^{\max}$ (Bonet and Geffner 2001), finding a disjunctive action landmark, and reducing the cost of these actions until the $h^{\max}$-value becomes zero. A precondition choice function (pcf) maps each action to a precondition with maximal $h^{\max}$-value. The pcf defines the $h^{\max}$ justification graph, where the facts of the task form the vertices and there are edges from the precondition returned by the pcf to each effect of an action. A disjunctive action landmark is constructed by computing a cut in the graph through actions leading into the zero-cost goal zone (nodes from which the goal can be reached with a zero-cost path), and the cost of these actions are reduced by the minimal action cost in that landmark.

Bonet and Helmert (2010) have shown that the heuristic is equal to $h^+$ if the landmarks are computed via hitting sets. They implemented variants of the heuristic using polynomial-time approximations of the hitting set and showed that this approach does improve the accuracy of the heuristic, but the improvement is outweighed by the added computational overhead.

However, there is also some room to make LM-cut more accurate without changing the algorithm: There can be multiple preconditions with the same (maximal) $h^{\max}$-value (in particular after the first few iterations when several actions have reduced costs), and their tie breaking is left unspecified. This detail has been mostly neglected in the literature, but as we demonstrate in our experiments, the performance of the heuristic varies a lot depending on the tie breaking strategy. Bonet et al. (Bonet and Helmert 2010; Bonet and Castillo 2011) introduced a variant with random tie breaking, where the heuristic is computed multiple times in each state and the maximum heuristic value is used. While the estimates of the heuristic values improve, requiring repeated computation of the heuristic diminishes its practical use. We explore several new tie breaking strategies that aim to improve the heuristic by generating more effective landmarks.

LM-cut provides accurate admissible estimates, but is expensive to calculate. Pommerening and Helmert (2012; 2013) show that the computational effort can be reduced through incremental computation, by caching landmarks and re-using them for the successor states. We introduce a new idea to speed up the LM-cut computation, by computing the cuts in the justification graph in a simplified and faster way. While this can lead to overapproximated cuts and make the heuristic less informed, this drawback is greatly outweighed by the improved computation speed.

We first summarize the general planning background and recapitulate the details of the LM-cut algorithm. Next, we describe and evaluate our optimization to the computation of the cuts. Finally, we introduce our new tie breaking strategies for the precondition choice function, explain the intuition behind them, and analyze their effectiveness with an empirical evaluation on the IPC benchmarks.

## Background

We first introduce the necessary background and notations, before reviewing the details of the LM-cut algorithm.

## Preliminaries

We consider classical planning using the STRIPS representation with action costs (Fikes and Nilsson 1971). A planning task is a 5-tuple $\Pi = (\mathcal{F}, \mathcal{A}, c, \mathcal{I}, \mathcal{G})$, where

- $\mathcal{F}$ is a finite set of *facts*,

- $\mathcal{A}$ is a set of actions, each $a \in \mathcal{A}$ is a triple of fact sets *preconditions* ($\text{pre}_a$), *add effects* ($\text{add}_a$), and *delete effects* ($\text{del}_a$) with $\text{add}_a \cap \text{del}_a = \emptyset$,

- $c$ is a cost function $\mathcal{A} \mapsto \mathbb{R}_0^+$,

- $\mathcal{I} \subseteq \mathcal{F}$ is the *initial state*,

- $\mathcal{G} \subseteq \mathcal{F}$ are the *goal facts*.

A *state* $s \subseteq \mathcal{F}$ is a set of facts. An action $a \in \mathcal{A}$ is *applicable* in a state $s$ if $\text{pre}_a \subseteq s$, and applying $a$ in $s$ leads to the state $s[\![a]\!] := (s \setminus \text{del}_a) \cup \text{add}_a$. A *plan* for $s$ is a successively applicable action sequence leading from $s$ to a goal state $s^* \supseteq \mathcal{G}$, and is called *optimal* if it has minimal cost among all plans for $s$. A plan for $\Pi$ is a plan for its initial state $\mathcal{I}$.

The set of all states is denoted by $\mathcal{S}$. A *heuristic function* (short *heuristic*) $h : \mathcal{S} \mapsto \mathbb{R}_0^+ \cup \infty$ estimates the cost of a plan for a state. The *perfect heuristic* $h^*$ returns the cost of an optimal plan. A heuristic $h$ is called *admissible* if $h(s) \leq h^*(s)$ for all $s \in \mathcal{S}$, and A* (Hart, Nilsson, and Raphael 1968) is guaranteed to find optimal solutions when using an admissible heuristic.

## LM-cut

LM-cut (Helmert and Domshlak 2009) is based on an iterative computation of (action) landmarks. A *disjunctive action landmark* is a set of actions $L$, such that every plan must include at least one action from $L$. In the following, we assume that $\mathcal{I} = \{i\}$, $\mathcal{G} = \{g\}$, and $|\text{pre}_a| \geq 1$ for all actions $a \in \mathcal{A}$ (this can be achieved with simple transformations). The heuristic is computed in an iterative procedure, where each iteration performs the following steps:

1. Compute $h^{\text{max}}$ (Bonet and Geffner 2001) for all facts. If $h^{\text{max}}(g) = \infty$, return $\infty$. If $h^{\text{max}}(g) = 0$, return the computed heuristic value.

2. Define a *precondition choice function* (pcf), mapping each action to a precondition with maximal $h^{\text{max}}$-value.

3. Construct the *justification graph*. The facts $\mathcal{F}$ of the planning task are the vertices of the graph, and there is an arc from the precondition chosen by the pcf to each of its add effects for all actions. The arc is labeled with the action.

4. Partition the vertices into three sets: (a) the 0-cost goal zone $V^*$, containing all facts from which $g$ is reachable with a 0-cost path, (b) the before-goal zone $V^0$, containing all facts reachable from $i$ without passing through a node in $V^*$, and (c) all other vertices $V^b$. The labels of the arcs leading from $V^0$ into $V^*$ define a disjunctive action landmark $L$.

5. Reduce the cost of actions in $L$ by $c_{\text{min}} = \min_{a \in L} c(a)$, and add $c_{\text{min}}$ to the heuristic value (which starts at 0).

After the computation of the heuristic, the action costs are reset to their original values.

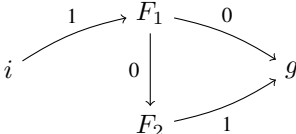

Figure 1: Example justification graph. A minimal cut only contains the action leading from $i$ to $F_1$.

Our work addresses two aspects of LM-cut. In the next section, we describe a faster method to compute the cut (step 4). Afterwards, we introduce and evaluate tie breaking strategies for the precondition choice function (step 2).

## Quick Cutting

In each iteration of LM-cut, the justification graph is built during the computation of $h^{\text{max}}$: whenever a fact is assigned its final $h^{\text{max}}$-value, (one of) its most expensive preconditions is stored as the predecessor in the justification graph. Afterwards, the zero-cost goal zone $V^*$ is constructed using a backwards exploration from $g$ considering only edges with cost 0. Finding the before-goal zone $V^0$ requires a separate forward exploration from $i$ considering only arcs that do not lead into $V^*$, to ensure that all vertices found by this procedure are reachable without passing through $V^*$.

We suggest to skip the forward exploration phase, and instead just consider all non-zero cost arcs that lead from a vertex not in $V^*$ to a vertex in $V^*$ for the cut. These arcs can easily be identified at the end of the backward exploration phase, and iterating over those is generally much cheaper than the forward exploration. However, this strategy may overapproximate the cut, as for some of the actions in the cut, the precondition selected by the pcf may not be reachable from $i$ without passing through the zero-cost zone.

Consider the example justification graph in Figure 1. The graph can be partitioned into $V^* = \{g, F_1\}$, $V^0 = \{i\}$, and $V^b = \{F_2\}$. According to the original algorithm, this would result in the cut containing only the arc from $i$ to $F_1$, whereas with our method, the cut would additionally include the one from $F_2$ to $g$. In this example, the heuristic value would not be affected, but the heuristic value can potentially change if the additional action would otherwise be included in a cut in a later iteration of the LM-cut computation.

Note that, since our computation of the cut always contains the cut as computed by the original algorithm (but may potentially include more actions), it is still a disjunctive action landmark, and does not affect the properties of the heuristic (in particular admissibility).

## Tie Breaking Strategies for LM-cut

The precondition choice function maps actions to a precondition with maximal $h^{\text{max}}$-value, but in many cases there are multiple such preconditions which leaves room for tie breaking. Consider the justification graph shown in Figure 2. The action achieving $g$ has three preconditions with maximal $h^{\text{max}}$-values: $v_1$, $v_2$, or $v_3$ (indicated by the dashed arrows). If $v_3$ is selected by the precondition choice function,

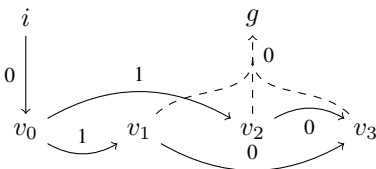

Figure 2: Example justification graph. The precondition choice function for the action achieving $g$ can select any of $v_1$, $v_2$, or $v_3$.

then the cut will contain both remaining actions with cost one, and the computation will terminate. However, if $v_1$ or $v_2$ is selected, the cut only contains one of these actions, and an additional cut can be made afterwards, increasing the heuristic value. The example is inspired by the VisitAll domain, where a perfect tie breaking strategy will only select actions leading to a single location for the cut in each iteration (making $h^{\text{LM-cut}} = h^+$). Different tie breaking may result in significantly larger cuts (and thus smaller heuristic values), and similar cases where tie breaking is important appear on most other domains as well.

In the following, we first explain how tie breaking is applied in detail, and then introduce several tie breaking strategies that aim to improve the heuristic.

## Tie Breaking in LM-cut

The pseudo code of the computation of the cut is shown in Algorithm 1. This shows our quick cutting method which only performs the backwards exploration from the goal; the original computation of the cut would perform the forward exploration afterwards to compute $V^0$, and would return $\{a \in Discovered \mid pcf(a) \in V^0\}$ instead.

It is important to note that for an action $a$, the tie breaking of the precondition choice function is only applied the first time $pcf(a)$ is evaluated — all subsequent calls to $pcf(a)$ will return the same selected precondition. Furthermore, at the time of the tie breaking, only a fragment of the final zero-cost goal zone $V^*$ is known.

Our tie breaking methods aim to reduce the size of the cuts, i.e., generating smaller disjunctive action landmarks in

---

**Algorithm 1:** $L$ computation

---
1   $V^* \leftarrow \{g\}$, $Explored \leftarrow \emptyset$, $Discovered \leftarrow \emptyset$
2   **while** $Explored \neq V^*$ **do**
3      select $f \in V^* \setminus Explored$
4      $Explored \leftarrow Explored \cup \{f\}$
5      **for each** $a \in \mathcal{A}$ with $f \in \text{eff}_a$ and
        $h^{\max}(p) < \infty$ for all $p \in \text{pre}_a$ **do**
6         **if** $c(a) = 0$ **then**
7            $V^* \leftarrow V^* \cup \{pcf(a)\}$
8         **else**
9            $Discovered \leftarrow Discovered \cup \{a\}$
10   **return** $\{a \in Discovered \mid pcf(a) \notin V^*\}$

---

step 4 of each LM-cut iteration. While this does not guarantee that the heuristic values will improve, it is reasonable to assume that reducing the cost of fewer actions will result in more cuts being made before $h^{\max}$ evaluates to zero.

In the following, we refer to the precondition selected by $pcf(a)$ as the *supporter* of $a$. We call an action $a$ an *achiever* of a fact $f$ if $f \in \text{eff}_a$. Our first tie breaking strategies aim to reduce the size of $V^*$, with the assumption that this leads to fewer actions pointing into $V^*$ and thereby smaller cuts.

**$V^*$ Detection (GZD)**   Prefer a precondition that is already in the zero-cost goal zone $V^*$. If such a precondition exists, then choosing it as the supporter can not increase $V^*$. Note that any further tie breaking among multiple potential supporters that are already in $V^*$ has no effect.

**Border Detection (BD)**   Prefer a precondition that has no zero-cost achievers. If such a supporter is selected, the zero-cost goal zone will not expand beyond that fact, as there are no further zero-cost actions to choose from. In our motivational example (Figure 2), this strategy would prefer $v_1$ and $v_2$ over $v_3$ as intended, minimizing $V^*$.

**Zero-Cost Achievers (ZCA)**   Prefer a precondition with a minimal number of zero-cost achievers. This strategy is an extension of the previous one (BD), but imposes a ranking to the potential supporters if there is no precondition without zero-cost achievers. The idea is again to approximate how many additional facts may be added to $V^*$ when recursively exploring the supporter. However, some of the zero-cost achievers may originate from other facts already in $V^*$, and these achievers would be beneficial in keeping $V^*$ small, so this approximation may not always be accurate.

**Value Decrease Minimization (VDM)**   Prefer a precondition of which the $h^{\max}$-value since the first iteration decreased the least. While the previous two strategies aim to reduce the breadth of the backward exploration beyond the supporter, this strategy aims to reduce the depth. If the $h^{\max}$-value of the precondition $p$ is close to its value from the first iteration, then there should not be many zero-cost actions in the justification graph between $i$ and $p$. In our example (Figure 2), assuming that the actions leading from $v_1$ to $v_3$ and $v_2$ to $v_3$ initially had a cost of 1, then this strategy would also prefer $v_1$ or $v_2$ over $v_3$.

**Zero-Cost Path (ZCP)**   Prefer a precondition $p$ that minimizes the number of zero-cost actions on a path from $i$ to $p$. Note that when the precondition choice function is called, the justification graph is still being constructed backwards from the goal so we do not know the structure of the graph between $i$ and $p$. Therefore, we partially re-use the justification graph from the previous iteration. More specifically, we keep track of the path information for each fact during the $h^{\max}$ computation: When the $h^{\max}$-value of a fact is set, we consider the justification graph from the previous LM-cut iteration including the incremental updates (with arbitrary tie breaking) of the current $h^{\max}$ computation so far. For the evaluation of the precondition choice function, we then select the candidate with the fewest zero-cost actions on its path. Like VDM, this strategy aims to reduce the depth of the

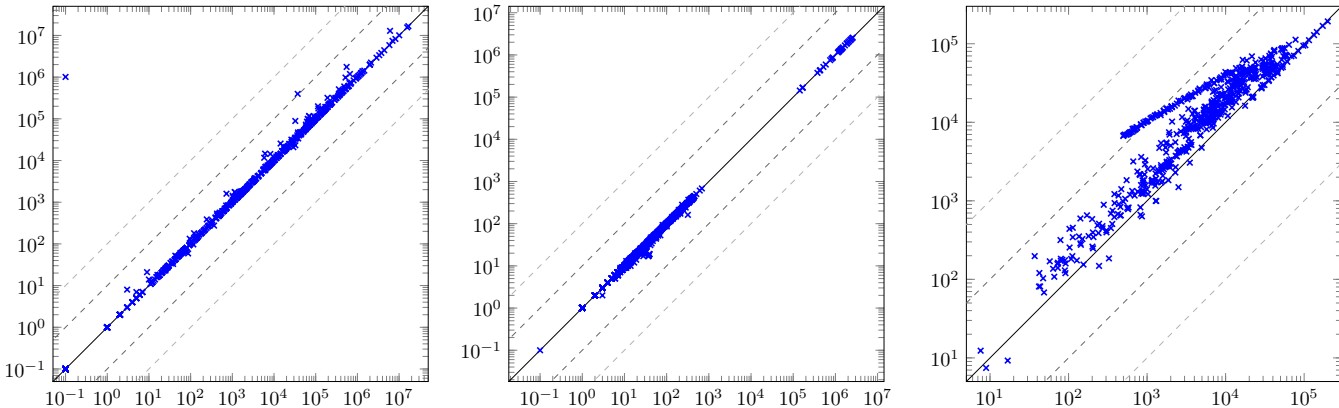

Figure 3: Number of expansions before the last $f$-layer (left), initial heuristic value (middle), and evaluations per second (right) for original (x-axis) vs. quick (y-axis) cutting. Expansions and evaluations per second are shown for commonly solved instances, and instances with a search time of less than $0.01$ seconds are excluded for the latter to reduce noise.

backward exploration by reducing the number of zero-cost actions between this precondition and $i$, but uses a different approximation.

**Achiever Minimization (AM)**  Prefer a precondition that has a minimal number of achievers with reachable preconditions under $h^{\text{max}}$. Each achiever of a fact $f$ adds an incoming arc to the corresponding node in the justification graph. Preferring preconditions with fewer achievers leads to fewer iterations in the backwards exploration (Algorithm 1, line 5). Thus, we can expect $V^*$ and *Discovered* to stay smaller, which should lead to fewer actions in the cut.

## Experiments

We implemented our techniques in Fast Downward (Helmert 2006), on top of the existing implementation of LM-cut. For our evaluation, we include all solvable instances from the optimal tracks of the IPCs up to 2018 that do not have conditional effects or axioms, resulting in a total of 1672 unique instances from 48 domains. The experiments were run using the Lab framework (Seipp et al. 2017) on a cluster of machines with Intel Xeon E5-2660 CPUs with a clock rate of 2.2 GHz. LM-cut is run in A* with time and memory limits of 30 minutes and 4 GB respectively. We first assess the impact of our alternative computation of the cut, before evaluating our new tie breaking strategies.

## Quick Cutting

Our new method of computing the cut should improve the computational efficiency of the heuristic, but at a potential loss of informativeness.

The left plot of Figure 3 shows a comparison of the number of expansions to the last $f$-layer between the two methods of computing the cut. Across all commonly solved instances, our method of computing the cut leads to $6.6\%$ more expansions on average. The domains where expansions increase the most are VisitAll ($+70\%$), Sokoban ($+40\%$), and Depot ($+23\%$), though the increase is typically low (less than $1\%$ on most domains). The initial heuristic value does

not decrease in 1396 of the 1672 instances (see the middle plot of Figure 3).

On the other hand, we obtain a huge speed-up in the computation of the heuristic as shown in the right plot of Figure 3. On average, with our method we can achieve $91\%$ more heuristic evaluations per second (up to $586\%$ in Miconic), which more than makes up for the comparatively small loss in informativeness. This leads to a decreased search time across almost all domains, and we are able to solve 28 more instances (856 vs. 828) on our benchmark set (the most significant gains are $+4$ in Parking and $+3$ in Floortile).

The single exception is Organic Synthesis (split). Since quick cutting overapproximates the cuts which leads to more actions having their cost reduced, intuitively, one would expect that this should lead to fewer iterations of $h^{\text{max}}$ per computation of $h^{\text{LM-cut}}$ on average. However, this is not always the case, as demonstrated by the following example. Assume there are two actions $a_1$ and $a_2$ with cost 2, which, using the original cutting, would be included in a cut at some point, having both of their costs reduced by 2. If instead $a_2$ was included in an earlier (overapproximated) cut which only reduced its cost by 1, then reducing both $a_1$ and $a_2$ to 0 may now require two cuts (first decreasing $a_1$ and $a_2$ by 1, and then reducing only $a_2$ by 1). Such cases appear frequently in Organic Synthesis. While the resulting heuristic value is typically not affected, it can significantly diminish the computational advantage of our cutting method. The coverage on Organic Synthesis does not decrease, but the number of evaluations per second drops by $18\%$, increasing the search time by $24\%$ on average.

## Tie Breaking Strategies

Table 1 shows the coverage of LM-cut with several tie breaking strategies. All configurations use our new method of computing the cuts (we also ran the experiments with the original cutting which exhibited a similar relative performance of the different tie breaking strategies). As baselines, we consider the (arbitrary) tie breaking employed by the cur-

| Coverage | arb | inv | rnd | GZD | BD | ZCA | VDM | ZCP | AM | GZD+BD | Scorpion |
|---|---|---|---|---|---|---|---|---|---|---|---|
| Airport (50) | 28 | 27 | 23 | **29** | 27 | 28 | 24 | 24 | 27 | 28 | **29** |
| Blocks (35) | **28** | 27 | **28** | 28 | 28 | 28 | 28 | 28 | 28 | 28 | 28 |
| DataNetwork (20) | 12 | 12 | 12 | 13 | 12 | 12 | 12 | 12 | 12 | 13 | **14** |
| Depot (22) | 7 | 7 | 7 | 7 | 7 | 7 | 7 | 7 | 7 | 10 | **13** |
| DriverLog (20) | 13 | 14 | 13 | 14 | 13 | 13 | 13 | 13 | 13 | 13 | **15** |
| Elevators (30) | 22 | 22 | 20 | 22 | 22 | 22 | 22 | 22 | 22 | 22 | **24** |
| Freecell (80) | 15 | 15 | 15 | 24 | 16 | 12 | 15 | 15 | 21 | 33 | **64** |
| Grid (5) | 2 | 2 | 1 | 2 | 2 | 2 | 2 | 2 | 2 | 2 | **3** |
| Hiking (20) | 10 | 10 | 9 | 10 | 9 | 8 | 9 | 9 | 10 | 9 | **14** |
| Logistics (63) | 27 | 27 | 25 | 27 | 25 | 25 | 25 | 25 | 25 | 27 | **34** |
| Mprime (35) | 23 | 25 | 22 | 23 | 23 | 23 | 22 | 22 | 25 | 24 | **31** |
| Mystery (19) | 16 | 17 | 15 | 17 | 17 | 17 | 17 | 17 | 17 | 17 | **19** |
| Nomystery (20) | 16 | 16 | 14 | 17 | 15 | 14 | 18 | 18 | 16 | 18 | **20** |
| Openstacks (80) | 31 | 31 | 31 | 31 | 31 | 30 | 31 | 31 | 31 | 31 | **34** |
| OrgSynth-split (20) | **15** | **15** | 14 | **15** | 14 | 10 | **15** | **15** | **15** | **15** | 10 |
| Parcprinter (30) | 19 | 22 | 19 | 22 | 19 | 19 | 22 | 22 | 18 | 20 | **30** |
| Parking (40) | 9 | 9 | 6 | 9 | 10 | 10 | 12 | 12 | 8 | **13** | **13** |
| Pegsol (36) | **35** | 34 | 33 | **35** | 34 | 34 | **35** | 34 | 34 | **35** | **35** |
| Pipes-notank (50) | 18 | 18 | 17 | 18 | 17 | 17 | 18 | 18 | 17 | 18 | **25** |
| Pipes-tank (50) | 12 | 12 | 10 | 12 | 11 | 9 | 12 | 12 | 12 | 12 | **18** |
| PNetAlignment (20) | **9** | **9** | 7 | **9** | **9** | **9** | **9** | **9** | **9** | **9** | 0 |
| Rovers (40) | 9 | **11** | 9 | 9 | 9 | 9 | 9 | 9 | 9 | 9 | 9 |
| Satellite (36) | 8 | 12 | 7 | 8 | 14 | 13 | **15** | **15** | 10 | 14 | 8 |
| Scanalyzer (30) | 16 | 16 | 16 | 16 | 15 | 14 | 16 | 16 | 16 | 16 | **18** |
| Snake (20) | 6 | 6 | 4 | 6 | 4 | 4 | 6 | 6 | 6 | 7 | **13** |
| Sokoban (30) | **30** | 29 | **30** | **30** | **30** | **30** | **30** | **30** | **30** | **30** | **30** |
| Spider (20) | 11 | 11 | 9 | 12 | 11 | 9 | 11 | 11 | 10 | 12 | **15** |
| Termes (20) | 7 | 6 | 6 | 7 | 6 | 6 | 7 | 7 | 6 | 7 | **13** |
| Tidybot (40) | **23** | 22 | 20 | **23** | 20 | 15 | 22 | 22 | 22 | **23** | 22 |
| VisitAll (40) | 16 | 15 | 17 | 15 | **36** | **36** | **36** | **36** | 14 | **36** | 30 |
| Woodworking (30) | 19 | 22 | 19 | 19 | 20 | 22 | 20 | 20 | 22 | 20 | **30** |
| Zenotravel (20) | **13** | **13** | 12 | **13** | 12 | 12 | **13** | **13** | **13** | 12 | **13** |
| Others (601) | 331 | 331 | 331 | 331 | 331 | 331 | 331 | 331 | 331 | 331 | **346** |
| **Sum (1672)** | 856 | 865 | 821 | 873 | 869 | 850 | 884 | 883 | 858 | 914 | **1020** |

Table 1: Coverage for LM-cut with different tie breaking methods. Domains where coverage across tie breaking methods does not change are grouped to "Others".

rent implementation of Fast Downward ("arb")[1], its inverse ("inv"), and random tie breaking ("rnd"). The results for random tie breaking are averaged and rounded over 5 random seeds, though there is very little variance (coverage changes across different seeds on only 4 instances, and overall coverage was always between 820 and 822). We include results for all our tie breaking strategies as well as selected combinations thereof. Furthermore, we added Scorpion (Seipp 2018)[2] as a representative of the state of the art.

This first thing to note is that the performance of LM-cut is heavily dependent on the tie breaking strategy, as the overall coverage ranges from 821 (with random tie breaking) to 914 (with one of our combined methods). The biggest differ-

ences can be seen in Freecell and VisitAll: depending on the tie breaking, the initial heuristic value can change by over a factor of 3 (Freecell) respectively 5 (VisitAll), and coverage ranges from 12 to 33 respectively 14 to 36. Most of our introduced strategies outperform the baselines, and random tie breaking is particularly bad across most domains. Four of our tie breaking strategies (BD, ZCA, VDM, and ZCP) lead to significant gains on VisitAll (and similarly on Satellite), as these strategies effectively solve the issue described in our corresponding example (Figure 2). In Freecell on the other hand, our other two strategies (GZD and AM) work best, increasing coverage by 9 respectively 6 over the baselines.

**Combined Tie Breaking Strategies**  In case of remaining ties, multiple tie breaking strategies can be used in sequence to break the remaining ties. In preliminary experiments, we had most success with combining tie breaking strategies that complement each other. For example, using any other strat-

---

[1]While there is no explicit tie breaking, typically the first one according to Fast Downward's variable ordering is selected.

[2]We disabled the $h^2$ preprocessor (Alcázar and Torralba 2015) to make it more comparable to our planner which is not using it.

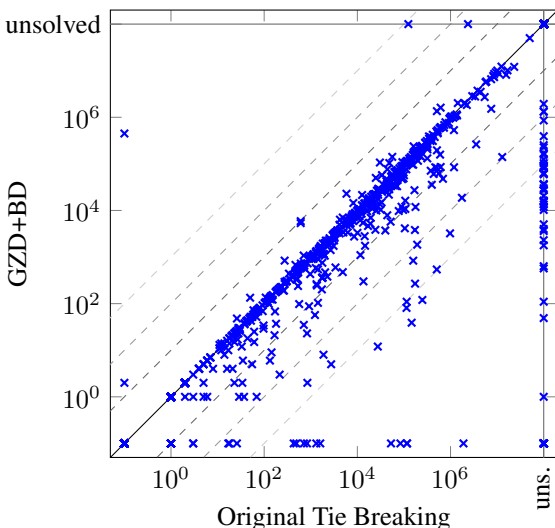

Figure 4: Number of expansions before the last $f$-layer for GZD+BD and original tie breaking.

egy after GZD or BD worked really well, as both strategies seem to make good tie breaking choices if their criterion applies, but yield no further information in case such a precondition does not exist. In that case, using a different strategy provides additional information on top, improving the performance of the heuristic. In contrast, VDM and ZCP both aim to reduce the depth of the backward exploration, and combining them had little effect.

The best configuration we have found so far prefers supporters that are already in $V^*$, and breaks the remaining ties according to BD (GZD+BD). This configuration retains the good performance on VisitAll, and further improves results on Freecell (+9 coverage compared to the best individual tie breaking method), Depot (+3), Parking (+1), and Snake (+1). Compared to the original tie breaking, coverage increases in 13 domains and decreases in only 2, solving 58 more instances overall. The number of expansions to the last $f$-layer decreases significantly (see Figure 4), sometimes by several orders of magnitude.

Compared to Scorpion, LM-cut is still worse on most domains. However, on Organic Synthesis (split) and Petri Net Alignment, the previous implementation of LM-cut already beat Scorpion; and with our improvements, LM-cut pulls ahead also in Sattelite (+7 coverage), VisitAll (+6), Rovers (+2), Tidybot (+1), and Miconic (+1).

## Conclusion

In this work, we introduced an optimization to LM-cut addressing the computation of the cut in the justification graph, and introduced and evaluated an extensive set of tie breaking strategies for the precondition choice function. Both contributions significantly improve the performance of LM-cut on the IPC benchmarks: our best performing configuration beats the previous implementation by 914 vs. 828 solved instances, increasing coverage by 86 using the same heuristic and search engine.

For future work, we want to explore additional tie breaking strategies, in particular ones that consider *which* actions should be included in the cut. Our current strategies only aim to include as few actions in the cut as possible, however, there may be cases where cutting more actions may be preferable (for example, if there is a single action that, if cut, enables a zero-cost path to the goal, but multiple other actions could be cut instead without enabling such a path).

Additionally, we want to analyze the combinations of multiple tie breaking strategies in more depth. In preliminary experiments, we had some good results, yet some combinations had surprisingly adverse effects on performance. Understanding the cause of these effects may allow us to combine our methods more effectively, and find combinations that may further improve the heuristic.

Furthermore, our methods seem complementary to incremental computation of LM-cut (Pommerening and Helmert 2013), and could be easily combined. This should further boost the performance of the heuristic, and make it more competitive with the state of the art.

## Acknowledgments

Maximilian Fickert was funded by DFG grant 389792660 as part of TRR 248 – CPEC (see https://perspicuous-computing.science).

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
