# OpenReview forum: "Beating LM-cut with LM-cut: Quick Cutting and Practical Tie Breaking for the Precondition Choice Function"
_icaps-conference.org/ICAPS/2020/Workshop/HSDIP — HSDIP 2020_

### Official Review · AnonReviewer2 · 2020-03-31
**Initial assessment**

**Rating:** 7
**Confidence:** 5

**Review:**

I think it is a nice paper with interesting results that provide more insight
into the LM-Cut heuristic.
However, I think the following (rather minor) issues should be addressed:
1. Bonet and Helmert (2010) is correctly cited as proposing to use maximum of
multiple passes of LM-Cut heuristics with different pcfs. But this, in my
opinion, is not the main point of that paper (it is actually mentioned only in
the experimental section as a variant for comparison). The main point (that
should be briefly discussed in this paper) is that computing a hitting set on the
collection of all landmarks for all possible pcfs will result in h^+ (which in
itself is a good theoretical justification for trying different pcfs). And they
propose to use polynomial approximation of this approach.

2. I don't like the way the word "efficiency" is used (both in text and in the
title). It suggest that the paper introduces a new method that allows to
compute the (qualitatively) same landmarks, but faster (hence it increases
efficiency). But what is proposed in the paper is to trade quality (or
informativeness) for faster evaluation (i.e., it (over)approximates the original
method). I think "efficient cutting" should be changed to "faster cutting" or
"approximate cutting" or something like that, because "efficient cutting" is,
in my opinion, misleading.

3. In footnote 1, it is said that the LM-Cut variants had similar results if
they used the original cutting, but in the "Efficient Cutting" section, it is
said that using the new cutting increases coverage by 28 problems. I'm not
sure what is meant by that footnote.
Moreover, I would like to see more results on the comparison between the
original and new cutting technique. It would be nice to see at least:
(a) a scatter plot of heuristic values for initial states for the "arb"
variant, and (b) an additional column in the table (it should be renamed from
Figure to Table) with the results for "arb" with the original cutting (which
is the "vanilla" LM-Cut variant). But, of course, the more interesting results
can fit into the paper, the better.

4. Why is h^2 disabled for Scorpion planner? Is it because h^2 is not used for
the LM-Cut variants? It is not clear from the text.

5. What is "inv" variant? It is inverse of what? Of "arb" -- what does it
mean?

6. I don't understand why the number of evaluations per second for the
organic-synthesis domain is smaller with the new cutting technique. I don't
think the large number of actions explains it, because there is the same
number of actions for both variants and it should still (at least) halve the
number of the exploration cycles (only V^* is explored, V^0 and V^b is not
computed at all).

---

> ### Comment · AnonReviewer2 · 2020-08-16
> **Good paper that should be accepted**
>
> The updated version of the paper seems to address all of my concerns, so I'm
> still for accepting the paper.
>
> However, I found one typo (below) and I have one question:
> In the "Quick Cutting" section you claim that the backward exploration phase
> "is generally much cheaper" than the forward exploration. If I'm not missing
> something, this is incorrect. Both backward and forward exploration require
> the same computational effort because they operate on the same graph. It is
> true that the backward exploration is typically cheaper in the first iteration
> because it needs to explore a smaller portion of the graph, but the forward
> exploration is cheaper in the last iteration for the same reason. In other words,
> by avoiding forward exploration, you are basically halving the time needed for
> finding the cut in the justification graph. Can you either fix this, or explain what
> am I missing?
>
> There is a typo in Preliminaries: add_a \cup del_a = \emptyset should be \cap

---

> > ### Author Response · Authors · 2020-09-04
> > **Author Response**
> >
> > Thanks again for the review.
> >
> > We believe there was a misunderstanding -- what we wanted to say is that considering the non-zero cost arcs into V* (i.e. our quick cutting method) is generally much cheaper than the forward exploration (i.e. the vanilla method). We acknowledge that this is currently not really clear from the text and can be interpreted as comparing the backward and forward exploration (which as you pointed out correctly should be similar in computational complexity), and will clarify the paragraph accordingly.
> >
> > Thanks for catching the typo!

---

### Official Review · AnonReviewer1 · 2020-03-31
**Initial assessment**

**Rating:** 8
**Confidence:** 4

**Review:**

The paper analyzes variations of the well-known LM-cut heuristic. The first alteration considers an approximation of the cut that considers *all* incoming arcs into the goal zone, rather than just those that can be reached without entering the goal zone. While it can weaken the heuristic, it speeds up the heuristic computation. The second alteration considers different tie-breaking strategies for the precondition choice function, aimed at minimizing the size of the cut.

Overall, the paper is well written, the intuition behind the different tie-breaking strategies is explained very well, and the experimental evaluation is thorough and presents interesting results.

There are some minor questions I had while reading and which I think would be interesting if the authors elaborate on them in the paper:

1) Figure 3 shows a single data point where traditional cutting resulted 0 (I assume) expansions before the last f-layer, but around 10^6 expansions with the new cutting. Could you elaborate on this? From which domain is this? Did traditional cutting result in the perfect heuristic, and how much worse was the new cutting to cause so many evaluations?

2) Figure 4 shows that for some problems efficient cutting resulted in over 10 times higher expansions but you say the highest increase is up to 586% more expansions in Miconic. Is that number the average for the Miconic domain, or how do these numbers add up?

3) I am not sure I can follow the argument why in Synthesis the new cutting is slower. Is it that for traditional cutting, the actual cut is very small, i.e. many actions in the new cut can only be reached by going through the goal zone?

4) For the comparison against Scorpion, why was the h^2 preprocessor disabled? Did you want a "pure heuristic comparison", i.e. that both planners perform standard A*, just with different heuristics?

---

### Author Response · Authors · 2020-04-17
**Author Response**

Thanks for the initial reviews. Given the overlap between your questions, we write a consolidated reply addressing both reviews.

Common Questions:

Regarding the cutting performance on Organic Synthesis:

In Organic Synthesis, there are lots of parallel edges between some facts (in some cases more than 100) in the justification graph. This structure does not appear in other domains, and causes some parts of the algorithm to have a more significant impact on the run time than in other domains. For example:
- If the cut is overapproximated using our new cutting method and contains significantly more actions, the associated bookkeeping/updating may affect run time.
- The "efficient cutting" configuration includes extensions for some of our tie breaking strategies. While these typically have negligible impact on the run time, in some domains this is more pronounced. We will re-run the experiments without these extensions in the final version.
However, in the meantime, we looked at a few instances in detail, and found that the number of h^max iterations can sometimes increase due to the overapproximated cuts (while the heuristic value remains unchanged). Consider the following example. There are two actions a1 and a2 with cost 2, which, using the original cutting, would be included in a cut at some point and have their cost reduced by 2. If a1 was included in an earlier (overapproximated) cut which reduced its cost by 1, then reducing both a1 and a2 to 0 may now require two cuts (first decreasing a1 and a2 by 1, and then reducing a2 by 1). This happens frequently in Organic Synthesis (usually without affecting the heuristic value), and is the main reason for the decrease in evaluations per second. We will clarify this in the paper.

Regarding the h^2 preprocessor:

Our planner is not using the h^2 preprocessor. In order to make a fair comparison, it seemed easier to disable it in Scorpion than integrating it into our planner.


Other Questions:

Reviewer 1:

1. The data point you described corresponds to an instance of Woodworking, with 0 vs. ~10^7 expansions to the last f-layer. The number of expansions to the goal is much closer (~7*10^6 vs. ~10^7), so while the heuristic is perfect on the initial state with the original cutting, this is not the case for most other states in that search space.

2. Yes, 586% is the average percentage increase on Miconic.


Reviewer 2:

1. Yes, we just wanted to point out that this is the only related work that experimented with different pcfs. We will add a brief discussion regarding the main point of the paper as an additional motivation.

2. We acknowledge that the word 'efficient' may be misleading, and will consider alternative naming options for the final submission.

3. The footnote should explain that the relative performance when using different tie breaking strategies is not affected by the choice of the cutting method, we will clarify this.

5. The "inv" variant is the inverse of "arb". Given a set of options to choose from, "arb" will pick the first one, whereas "inv" will pick the last one.

---

### Comment · AnonReviewer1 · 2020-08-14
**Comments to revised version**

The revised version addresses almost all comments from the initial assessment
(point 3 from Reviewer 2 is not fully addressed) and I'm happy with the
changes, so still a clear accept from my side.

Minor remarks:
 - ZCP tie breaking: could you elaborate how pcf' is computed? Does it take the
 partial current iteration (or previous ones) into account?
 - Preliminaries: cost function A \mapsto ... -> \mathcal A
 - Footnote 1: What does "typically" mean here? Are there cases where tie
 breaking does not follow Fast Downward's variable order?
 - References: Franco et al. 2017 misses page numbers

---

> ### Author Response · Authors · 2020-09-04
> **Author Response**
>
> Thanks again for the review.
>
> Arbitrary tie breaking:
> The tie breaking follows Fast Downward's variable ordering in all but the first iteration of the LM-cut computation. More specifically, in the initial h^max computation, the precondition achieved last in that process is set as the supporter for each action. If multiple preconditions have the same h^max value, the order in which they are removed from the queue (and hence their tie breaking for the pcf) is not specified, and depends on internals of the priority queue. Whenever the h^max value of a fact is updated (decreased) in subsequent iterations, the supporter of all actions where this fact was previously selected as the supporter is reset and updated. In the updating process, all preconditions of the action are considered in order (corresponding to FD's variable order), and the first one with minimal h^max value is selected.
>
> The "inv" tie breaking works in the same way, but inverts the iteration order when updating the supporter.
> Note that these two tie breaking strategies work differently from the other tie breaking strategies: while the choices for "arb" and "inv" are fixed during the h^max computation, the other tie breaking strategies are evaluated in the backward exploration phase as shown in Algorithm 1.
>
>
> ZCP tie breaking:
> For each fact f, we keep track of the minimal number of 0-cost actions on any path from i to f during the h^max computation. These paths are considered on a justification graph that is carried over from the previous iteration, including incremental updates of the current h^max computation so far (with arbitrary tie breaking). The information about the number of 0-cost actions along these paths is then used to construct the new justification graph in the backward phase. We will also update its description in the paper.
>
> Thanks for catching the typo and the missing page numbers!

---

### Comment · Program_Chairs · 2020-09-14
**Final Decision: Accept**

Dear Authors,

Thank you very much for your submission. We are happy to inform you that we have decided to accept it and we look forward to your talk in the workshop. You will receive additional information per mail in the coming days.

Best,
The HSDIP'20 team

---

### Decision · Program_Chairs · 2020-09-30

Accept